

# Membrane and luminal proteins reach the apicoplast by different trafficking pathways in the malaria parasite *Plasmodium falciparum*

Rahul Chaudhari[1], Vishakha Dey[2], Aishwarya Narayan[2], Shobhona Sharma[1] and Swati Patankar[2]

[1] Department of Biological Sciences, Tata Institute of Fundamental Research, Mumbai, Maharashtra, India
[2] Department of Biosciences and Bioengineering, Indian Institute of Technology, Bombay, Mumbai, Maharashtra, India

## ABSTRACT

The secretory pathway in *Plasmodium falciparum* has evolved to transport proteins to the host cell membrane and to an endosymbiotic organelle, the apicoplast. The latter can occur via the ER or the ER-Golgi route. Here, we study these three routes using proteins Erythrocyte Membrane Protein-1 (PfEMP1), Acyl Carrier Protein (ACP) and glutathione peroxidase-like thioredoxin peroxidase (PfTPx$_{Gl}$) and inhibitors of vesicular transport. As expected, the G protein-dependent vesicular fusion inhibitor AlF$_{4-}$ and microtubule destabilizing drug vinblastine block the trafficking of PfEMP-1, a protein secreted to the host cell membrane. However, while both PfTPx$_{Gl}$ and ACP are targeted to the apicoplast, only ACP trafficking remains unaffected by these treatments. This implies that G protein-dependent vesicles do not play a role in classical apicoplast protein targeting. Unlike the soluble protein ACP, we show that PfTPx$_{Gl}$ is localized to the outermost membrane of the apicoplast. Thus, the parasite apicoplast acquires proteins via two different pathways: first, the vesicular trafficking pathway appears to handle not only secretory proteins, but an apicoplast membrane protein, PfTPx$_{Gl}$; second, trafficking of apicoplast luminal proteins appear to be independent of G protein-coupled vesicles.

## INTRODUCTION

*Plasmodium falciparum* parasites export proteins to the plasma membrane of host erythrocytes, cells that do not possess their own trafficking machinery. In order to do so, the parasite extensively modifies the host cell to make a favorable niche for survival (*Moxon, Grau & Craig, 2011*). The parasite can, therefore, be considered a major secretory cell.

In the secretory pathway, proteins are targeted to their destinations by the endomembrane system, starting with the proteins' entry into the endoplasmic reticulum (ER), a process facilitated by N-terminal signal sequences that are usually hydrophobic in nature. From the ER, proteins are sent to the Golgi and further to their final destinations. In *P. falciparum*, the ER consists of a tubular, interconnected network that surrounds the nucleus, while the

Corresponding author
Swati Patankar, patankar@iitb.ac.in

unstacked Golgi apparatus consists of distinct cis- and trans- compartments (*Struck et al., 2005*; *Van Dooren et al., 2005*). In mammalian cells, the position and integrity of the ER and Golgi are maintained by microtubules, which also act as tracks for vesicles that target proteins via the secretory pathway (*Cole & Lippincott-Schwartz, 1995*).

Once inside the ER, the proteins navigate different paths according to their targeting signals and destinations (*Deponte et al., 2012*). For example, *P. falciparum* Erythrocyte Membrane Protein-1 (PfEMP-1) has N-terminal transmembrane regions which act as signal sequences, sending the protein via the secretory route to the parasite plasma membrane from where they are exported to the host cell surface (*Knuepfer et al., 2005*). In addition to export, proteins are also trafficked internally to parasite subcellular compartments, including an unusual relict plastid, the apicoplast. The apicoplast is believed to be acquired by secondary endosymbiosis and is surrounded by four lipid bilayers (*Lemgruber et al., 2013*; *McFadden & Roos, 1999*). The organelle possesses a 35 kb circular genome that codes for a handful of housekeeping genes and, as a result, is heavily dependent on the import of nuclear-encoded proteins (*Marechal & Cesbron-Delauw, 2001*).

A protein destined for the apicoplast lumen is marked by an N-terminal bipartite signal, comprising of a signal peptide, for entry into the secretory pathway at the ER, and a transit peptide, required for luminal import by translocons upon reaching the apicoplast (*Tonkin et al., 2006b*; *Waller et al., 2000*). Once inside, the transit sequence is removed by an organellar peptidase to form a mature functional protein (*Van Dooren et al., 2002*).

Since proteins that enter the ER usually follow the secretory route, the trafficking of a luminal protein from the ER to the apicoplast might be expected to go via the Golgi. However, in *P. falciparum*, there are two models for trafficking of apicoplast proteins from the ER lumen. In one report, it is suggested that a luminal protein Acyl Carrier Protein fused to Green Fluorescent Protein (ACP-GFP) is transferred from the ER to the apicoplast bypassing the Golgi (*Tonkin et al., 2006b*). This has been hypothesized to occur directly via vesicles, or due to transient contacts between the membranes of the two organelles. Interestingly, in another study ACP-GFP was suggested to transit through the Golgi (*Heiny et al., 2014*); reconciling these two reports remains an open area.

Another apicoplast protein, the glutathione peroxidase-like thioredoxin peroxidase (PfTPx$_{Gl}$) of *P. falciparum* localizes to the apicoplast and/or mitochondrion. This heterogeneous localization of PfTPx$_{Gl}$ is completely disrupted upon BFA treatment suggesting an ER-Golgi route for organellar localization (*Chaudhari, Narayan & Patankar, 2012*). In contrast to ACP, its targeting does not involve the cleavage of N-terminal signal sequences. Another group has localized this protein to the apicoplast and the cytosol by fusion of N-terminal 47 amino acids to GFP (*Kehr et al., 2010*).

Clearly, in *P. falciparum*, once proteins enter the ER, they could have different fates. These include export via the Golgi and secretory pathway, trafficking to the apicoplast via the Golgi and trafficking to the apicoplast directly from the ER. In this report, we study the trafficking of two apicoplast proteins (PfTPx$_{Gl}$ and ACP). Inhibition of vesicular fusion and transport were carried out using aluminum tetrafluoride (AlF$_4^-$), a small molecule inhibitor of vesicle fusion to target membranes and vinblastine, a microtubule depolymerizing agent that disrupts vesicular transport. These inhibitors have been well characterized

in *Plasmodium falciparum* and shown to target the same functions as in other eukaryotes (*Chakrabarti et al., 2013*; *Taraschi et al., 2001*). PfTPx$_{Gl}$ localization is disrupted by AlF$_4$– and vinblastine while the localization of luminal apicoplast proteins (including ACP) is unaffected by the same concentrations of these compounds, suggesting that PfTPx$_{Gl}$ and ACP trafficking proceeds by two different routes. The nature of the signals on these proteins and the signals on different types of vesicles that dictate the choice of the trafficking routes emanating from the ER is now an avenue for future research. One such signal to direct apicoplast proteins through the Golgi could be membrane localization: here we show that PfTPx$_{Gl}$ isassociated with the outermost membrane of apicoplasts, suggesting that, unlike luminal proteins, the protein is trafficked on vesicular membranes.

## MATERIALS & METHODS

### Ethical clearance

The work was approved by the Institute ethics committee and Institute biosafety committee at Indian Institute of Technology Bombay. Written informed consent was provided by all the blood donors.

### *In vitro* culture of *P. falciparum* erythrocytic stages

For parasite culture, blood was collected from healthy donors who provided written informed consent. *P. falciparum* 3D7 strain was cultured in RPMI 1640 (Gibco, Waltham, MA, USA) with an additional 2 mg ml$^{-1}$ sodium bicarbonate (Sigma-Aldrich, St. Louis, MO, USA), supplemented with 10% B+ human plasma, 48 mg L$^{-1}$ hypoxanthine (Sigma-Aldrich, St. Louis, MO, USA), 2 mg ml$^{-1}$ glucose (Sigma-Aldrich, St. Louis, MO, USA) and 50 µg ml$^{-1}$ gentamicin (Abbott, Chicago, IL, USA). A hematocrit of 3% was maintained using human B$^+$ red blood cells. Parasites were tightly synchronized for all experiments except for the Western blot experiment showing membrane extraction. Briefly, the parasites were synchronized at the early rings stage (4–5 h post infection) by treatment with 5% sorbitol (10 volumes of the RBC pellet) for 10 min at 37 °C followed by two washes with incomplete RPMI. Resulting pellet was suspended in complete medium as described above. The treatment was repeated after 4 h to synchronize the cultures tightly. The synchronization was confirmed by observing smears prepared from the synchronized cultures.

### Immunofluorescence microscopy

Immunofluorescence microscopy of *P. falciparum* D10-ACP$_{leader}$-GFP parasites was done as described earlier with slight modifications (*Tonkin et al., 2004*). All steps up to incubation with secondary antibodies were performed according to *Tonkin et al. (2004)*. The details about combinations and dilutions of the antibodies and their method of generation and checking the specificity are shown in Table S1. Primary antibodies treatment was performed overnight at 4 °C for all the proteins and secondary antibodies treatment was performed for 1.5 h at room temperature.

After incubation with secondary antibodies, the cells were subjected to three PBS washes in suspension and allowed to settle on poly-L-lysine (Sigma-Aldrich, St. Louis,

MO, USA) coated cover slips for 10 min. Cover slips were washed thrice in PBS, air dried and mounted using 0.1 mg ml$^{-1}$ 1,4-diazylbicyclo[2.2.2]octane (DABCO; Sigma-Aldrich, St. Louis, MO, USA) on to glass slides. Slides were examined with Olympus® FluoView® 500 Confocal Laser Scanning Microscope. For each experiment consisting of a control and treated cultures, all images were acquired at identical settings (laser power ranging from 0.5 to 4%). For clarity, the images were processed later for brightness and contrast using ImageJ 1.46r where adjustments were applied to whole image. No non-linear adjustments were performed.

### Treatment of *Plasmodium falciparum* with inhibitors of vesicular trafficking

Small molecule inhibitors were used to disrupt the vesicular trafficking pathway. Small molecules can have pleiotropic effects; therefore, drugs that affect different steps of vesicular transport were chosen with the expectation of obtaining consistent results with all the treatments. Further, the same treatments are tested for their effects on three different proteins, in an attempt to dissect out the pathways used by these proteins under the identical conditions.

For all drug treatments, it was important to ensure that the observed signal was not from previously accumulated organellar protein. This would require treating the parasites with drugs for a period of time that would be close to or greater than the half-life of the protein being studied. The most stable protein studied here is likely to be GFP, whose half-life is estimated to be 26 h in mammalian cells (*Corish & Tyler-Smith, 1999*) however, can be as short as 2 h in some cells (*Halter et al., 2007*). Hence, rather than treating parasites with high concentrations of drugs for a few hours, as has been done previously (*Kaderi Kibria et al., 2015*; *Taraschi et al., 2001*), we first determined the IC$_{50}$ concentrations of the drugs and then treated parasites for 18–20 h with these lower concentrations (Fig. S1).

### I. IC$_{50}$ determination for the AlF$_4^-$ treatment

IC$_{50}$ determination for the AlF$_4^-$ treated parasites were performed in 24 well plates where the reduction in parasitemia at different AlF$_4^-$ concentrations was monitored. The experimental set-up included 2 ml of tightly synchronised *P. falciparum* 3D7 cultures grown as described previously in the materials and methods section. For these experiments, two plates (two biological replicates) were maintained containing two technical replicates for each AlF$_4^-$ concentration. For generation of AlF$_4^-$ (1 μM–10 μM), appropriate amounts of AlCl$_3$ (Sigma-Aldrich, St. Louis, MO, USA) (10 mM stock) and NaF (Sigma-Aldrich, St. Louis, MO, USA) (1M stock) were added. The plates, kept in duplicate, were maintained for two life cycles (96 h). Spent media was replaced with fresh media every 24 h with the addition of fresh AlCl$_3$ and NaF for the generation of AlF$_4^-$. Reduction in the parasitemia at different concentrations of AlF$_4^-$ was assessed with smears prepared using Field's stain every 24 h. Percentage parasitemia as compared to controls were calculated by subtracting the parasitemia of the AlF$_4^-$ treated cultures from the parasitemia of the control (without AlF$_4^-$) for each time point. The graph of percent parasitemia compared to the controls was plotted against drug concentrations and 50% inhibitory concentration (IC$_{50}$) was calculated with calculated with non-linear regression of the sigmoidal dose response equation from OriginPro for Windows.

## II. Aluminum tetrafluoride treatment

D10-ACP$_{leader}$-GFP and 3D7 parasites were treated with 1.2 μM AlF$_{4-}$ (IC$_{50}$ concentration). Briefly, for generation of AlF$_{4-}$, AlCl$_3$ (Sigma-Aldrich, St. Louis, MO, USA) and NaF (Sigma-Aldrich, St. Louis, MO, USA) were combined in 5 ml complete medium to a final concentration of 1.2 μM AlCl$_3$ and 0.36 mM NaF (AlCl$_3$–10 mM stock and NaF-1 M stock). A total of 150 μl of packed infected red blood cells (having parasitemia of 5% early rings) were then added to 1.2 μM AlF$_{4-}$ containing complete medium (final hematocrit of 3%). The cultures were incubated at 37 °C for 18 ± 2 h. The AlF$_{4-}$ treated cultures were then washed three times with incomplete medium and a final wash with PBS. This was followed by preparation of immunofluorescence slides as described previously.

## III. Treatment of the *P. falciparum* D10-ACP$_{leader}$-GFP parasites with microtubule destabilizing drugs

D10-ACP$_{leader}$-GFP parasites were treated with nocodazole (Sigma-Aldrich, St. Louis, MO, USA) and vinblastine (Sigma-Aldrich, St. Louis, MO, USA) at their IC$_{50}$ concentrations 17 μM and 100 nM, respectively, in a final volume of 5 ml. Vinblastine stock of 50 μM was prepared in phosphate buffered saline while a 10 mM nocodazole stock solution was prepared in DMSO. Both stock solutions were sterilized by passing through 0.2μ membrane PVDF filters (Merck-Millipore, Billerica, MA, USA). One culture flask was treated with an equal volume of DMSO (Sigma-Aldrich, St. Louis, MO, USA) as a control.

After 18 ± 2 h incubation with drugs, 2 ml of cultures were removed, washed three times with incomplete medium and a final wash with PBS and usedfor PfTPx$_{Gl}$ andmicrotubule staining. This was followed by preparation of immunofluorescence slides as described previously. For assessing the reversion of localization in drug washed out parasites, the remaining cultures were washed thrice with complete medium and resuspended in a complete medium without drugs and subjected to additional incubation of 4 h followed by preparation of immunofluorescence slides.

## Immunofluorescence microscopy of intact organelles

For detection of membrane localized PfTPx$_{Gl}$, intact organelles were isolated according to a previous report (*Mullin et al., 2006*) except the parasites were lysed by expulsion through a 26-gauge needle (20 times) and the organelles including the apicoplast in the post-nuclear fraction were then centrifuged at 13,000 g for 20 min, 4 °C. The organellar pellet was divided into two fractions.

To detect proteins located on the membranes of organelles, intact organelles were re-suspended in 1X assay buffer containing 1% BSA and shaken at 4 °C for 30 min for blocking. This was followed by incubation with anti-PfTPx$_{Gl}$ antibodies (1:100) and anti-GFP antibodies (1:250) for 4 h at 4 °C in separate reaction tubes. After one wash with 1X assay buffer containing 1% BSA, organelles were treated with secondary antibodies (goat anti-rabbit IgG (H + L) Alexa Fluor$^®$ 568 (Invitrogen$^{TM}$) diluted 1:200 for detection of anti-PfTPxGl antibodies and Goat anti-Mouse IgG (H + L) Alexa Fluor$^®$ 568 (Invitrogen$^{TM}$) (1:250) for detection of anti-GFP antibodies) in 1X assay buffer containing 1% BSA for 1 h at 4 °C. This was followed by a final wash with 1X assay buffer without 1% BSA and fixation

with 4% paraformaldehyde and 0.0075% glutaraldehyde for 30 min on ice. This prep was then treated with primary and secondary antibodies and the slides were prepared as described previously. Mitochondria in the isolated organelles were visualized by staining with Mitotracker Red CM-H$_2$XRos (Invitrogen$^{TM}$) at 50nM final concentration for 30 min at room temperature.

To detect proteins within the organelles, organelles were first fixed with 1X assay buffer containing 4% paraformaldehyde and 0.0075% glutaraldehyde for 30 min on ice. Organelles were then permeabilized with 0.1% Triton X-100 in 1X assay buffer for 10 min on ice. The preparation was blocked with 3% BSA, treated with primary and secondary antibodies and the slides were prepared as described above.

## Differential solubilization of PfTPx$_{Gl}$ and Western blotting

Approximately $6 \times 10^9$ parasites were lysed hypotonically in de-ionized water containing protease inhibitors followed by three rounds of freeze-thaw cycles. The resulting suspension was divided equally in three different parts. This was followed by centrifugation at 36,000 g for 30 min at 4 °C. The supernatant containing soluble proteins was removed and the pellets were subjected to 1% Triton X-100-PBS for 30 min at 4 °C. This was followed by centrifugation at 36,000 g for 30 min at 4 °C to obtain insoluble fraction and supernatant containing integral membrane proteins. Proteins were then quantified by bicinchoninic assay using BSA as a standard. 200 µg of each protein fraction (soluble proteins and integral membrane proteins extracted with Triton X-100) was then separated on 15% SDS-PAGE.

Proteins were transferred to polyvinylidenedifluoride (PVDF) membranes (pore size 0.45 µm; Millipore, Billerica, MA, USA). The membranes were blocked for an hour with 3% BSA/PBS. The membranes were then incubated for 3 h in 0.5% Tween-20 (Sigma-Aldrich, St. Louis, MO, USA)/PBS containing rabbit raised anti-PfTPx$_{Gl}$ serum at 1:2000 dilution and mouse anti-GFP antibodies (for detection of ACP-GFP) at 1:1000 dilution at room temperature. This was followed by three washes with PBS. The proteins were probed with horseradish peroxidase-conjugated goat anti-rabbit secondary antibodies (Merck Biosciences, Keniworth, NJ, USA) (1:2000) for 1.5 h. This was followed by three washes with PBS and detection of the protein bands with 1.6 mM 3, 3′-diaminobenzidinetetrahydrochloride (DABCO) and 0.1% hydrogen peroxide as substrates in 10 ml of 0.01M Tris (pH 7.6). Molecular size of the protein bands were determined with reference to pre-stained protein molecular weight markers (Fermentas, Waltham, MA, USA).

## Thermolysin treatment

Thermolysin treatment of isolated organelles was carried out as described previously with minor modifications (*Mullin et al., 2006*). Intact organelles were isolated as mentioned above however the hypotonic buffer did not contain EGTA and protease inhibitors. The 4× assay buffer used was 200 mM HEPES-NaOH (pH 7.4), 1.2 M sorbitol (Sigma-Aldrich, St. Louis, MO, USA) and 2 mM CaCl$_2$(Merck Biosciences, Kenilworth, NJ, USA). The organellar pellet was divided into six fractions. One fraction was used for protein estimation by Bradford assay using BSA as a standard. The remaining pellets were treated as follows.

(i) No thermolysin, (ii) 25 μg thermolysin (Sigma-Aldrich, St. Louis, MO, USA) per mg of parasite proteins, (iii) 25 μg thermolysin per mg of parasite proteins and 10 mM EDTA (to inhibit the thermolysin), (iv) 25 μg thermolysin per mg of parasite proteins and 1% Triton X-100 (to permeabilize the organelles), (v) 25 μg thermolysin per mg of parasite proteins, 1% Triton X-100 and 10 mM EDTA. After 30 min incubation at 30 °C, thermolysin was inhibited by adding EDTA to a final concentration of 10 mM. Protein were precipitated by chloroform/methanol/water and analyzed by Western blotting.

## RESULTS

### Aluminum Tetrafluoride (AlF$_4$−) disrupts localization of PfEMP-1, KAHRP and PfTPx$_{Gl}$, leaving ACP and PfUROD localization unaffected

Heterotrimeric G proteins control the recognition and fusion between transport vesicles and their acceptor compartments (*Balch, 1992*; *Takai, Sasaki & Matozaki, 2001*). AlF$_4$− binds to the Gα subunit of G proteins by mimicking the γ-phosphate group of GTP; as a result, the heterotrimeric G protein remains in an active state even after GTP is hydrolysed to GDP (*Chabre, 1990*; *Finazzi et al., 1994*; *Kahn, 1991*). This continuous activation inhibits ARF-mediated coatomer coat shedding from vesicles. The resulting inhibition of vesicle fusion with target membranes after treatment with AlF$_4$− has been demonstrated in several organisms, including *Plasmodium* (*Taraschi et al., 2001*). The majority of trafficking vesicles are inhibited by AlF$_4$−, the only exception so far being endocytosis of CD94/NKG2A in natural killer cells (*Masilamani et al., 2008*).

PfTPx$_{Gl}$ was shown to be trafficked to the apicoplast by a Brefeldin-A sensitive pathway which suggests transit through the Golgi (*Chaudhari, Narayan & Patankar, 2012*); these data further indicated a vesicular component for targeting of PfTPx$_{Gl}$. For another apicoplast protein, ACP, current models of trafficking have suggested that it may be transferred from the ER to the apicoplast via the Golgi (*Heiny et al., 2014*), from the ER to the apicoplast by vesicles, or by direct transfer due to transient contiguity between the membranes of the two organelles (*Tonkin et al., 2006b*). As AlF$_4$− inhibits vesicular fusion, these hypotheses were tested.

D10-ACP$_{leader}$-GFP parasites, in which GFP targeting to the apicoplast is shown to be independent of the Golgi (*Tonkin et al., 2006b*), were used for treatment with AlF$_4$− and localization of ACP-GFP and PfTPx$_{Gl}$ was analyzed. Parasites were treated with IC$_{50}$ concentration of AlF$_4$− (1.2 μM, Fig. S1A), which did not alter parasite morphology (Fig. S1B).

First, the efficacy of the AlF$_4$− treatment at the IC$_{50}$ concentration of 1.2 μM was analyzed by observing its effect on the secreted protein PfEMP1. Previously, treatment with 100 μM AlF$_4$− for two hours inhibited the fusion of PfEMP1 containing vesicles with target membranes (*Taraschi et al., 2001*). In this report, in parasites treated with 1.2 μM AlF$_4$− for 18 h, PfEMP1 trafficking to the host RBC was inhibited as a majority of the protein was observed in the parasite (Fig. 1B, Table S2C). This was in contrast to untreated parasites where the protein was found both in the parasite and in punctate structures in the host RBC (Fig. 1A). Similar results were obtained with another secretory protein KAHRP (Fig. S2, Table S2C), confirming that conditions for AlF$_4$− treatment used in this study are robust for secretory proteins.

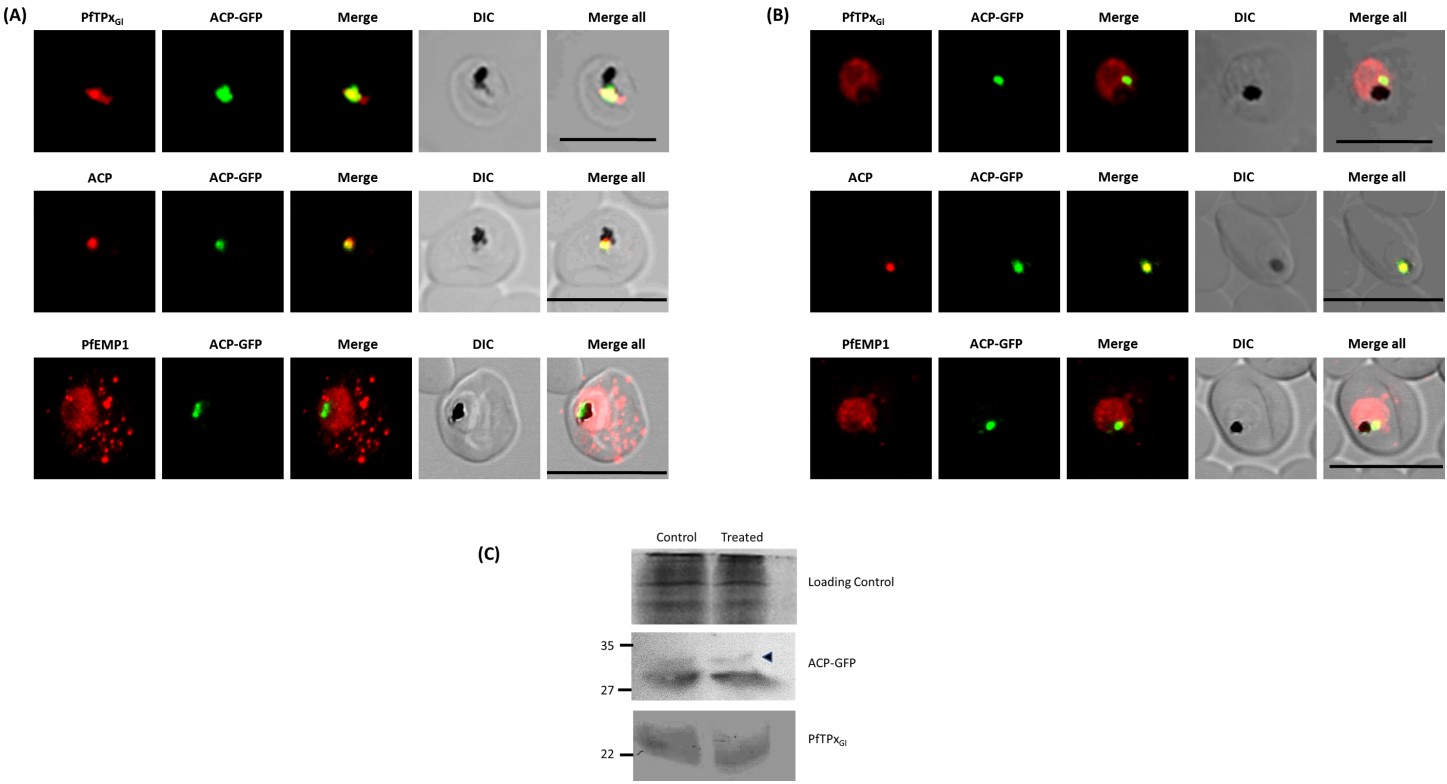

**Figure 1** **Immunofluorescence images show PfTPx$_{Gl}$, microtubules, ACP-GFP, PfACP and PfEMP1 trafficking in AlF$_4-$ -treated D10-ACPleader-GFP parasites.** (A) PfTPx$_{Gl}$, ACP-GFP, PfACP and PfEMP1 localization in control parasites, (B) PfTPx$_{Gl}$, ACP-GFP, PfACP and PfEMP1 localization in AlF$_4-$ -treated parasites. For D10-ACPleader-GFP parasites, 98% of the 115 parasites analyzed showed disrupted PfTPx$_{Gl}$ signal while 96% of the 147 parasites analyzed showed arrest of PfEMP1 in the parasites. Scale Bar: 10 μm, (C) Processing of apicoplast targeted protein ACP-GFP is not affected in drug-treated parasites. Western blot showing the processing of apicoplast targeted ACP-GFP and PfTPx$_{Gl}$ in AlF$_4-$ - treated parasites. In ACP-GFP panel, upper band indicated with an arrowhead represents an unprocessed form of ACP-GFP while lower band represents processed form of ACP-GFP.

Next, we studied the localization of PfTPx$_{Gl}$ upon treatment with AlF$_4-$. The control cultures grown in parallel to the treated cultures showed apicoplast localization of PfTPx$_{Gl}$ between 45% and 55% (Table S2A). However, in around 98% AlF4$^-$ treated parasites observed, PfTPx$_{Gl}$ targeting was disrupted and punctate staining was visible throughout the parasite (Fig. 1B). No co-localization was observed with an apicoplast marker protein for any of these parasites. This indicated that apicoplast localization is affected by the treatment. When the parasites from the same treated cultures were analyzed for co-localization of the disrupted PfTPx$_{Gl}$ signal with mitochondrial marker protein ferrochelatase (PfFC), we observed that in 35–40% of cells the PfTPx$_{Gl}$ signal showed some overlap (Fig. S3). This data indicated that trafficking of PfTPx$_{Gl}$ to the mitochondrion may be partially disrupted by the treatments and requires further characterization. In contrast, as the apicoplast signal was disrupted in 98% of parasites, we chose to focus on only the apicoplast localization of this protein.

In contrast, targeting of the apicoplast marker protein ACP-GFP was not disrupted and showed localization to distinct structures indicative of the organelle (Fig. 1B). To exclude

the possibility that these observations were an artifact of the GFP fusion of this protein, endogenous ACP was also monitored in AlF$_4$– treated parasites by immunofluorescence assays. In control as well as in treated parasites, endogenous ACP colocalized perfectly with ACP-GFP signal suggesting that it was unaltered by the treatment (Figs. 1A and 1B)

As import of ACP-GFP into the apicoplast leads to transit peptide cleavage which can be monitored by Western blot (*Van Dooren et al., 2002*), AlF$_4$– treated parasites (treated for $18 \pm 2$ h) were subjected to this analysis. Two bands were observed on the Western blot; the uppermost band represents the unprocessed form (indicated by an arrowhead) while the lower band represents the processed form of ACP-GFP (Fig. 1C) (*Waller et al., 2000*). The majority of ACP-GFP from AlF$_4$– treated parasites was observed to be in the processed form, corroborating the results obtained with immunofluorescence assays. As expected, PfTPx$_{Gl}$ was not processed as reported earlier (*Chaudhari, Narayan & Patankar, 2012*).

Similar experiments were done in the 3D7 parasite strain with another luminal apicoplast protein Uroporphyrinogen III decarboxylase (PfUROD). Antibodies recognizing PfUROD have been previously characterized (*Nagaraj et al., 2009*). Here, parasites treated with 1.2 µM AlF$_4$– for 18 h showed PfTPx$_{Gl}$ distributed throughout the cell, while the targeting of PfUROD was not disrupted (Fig. S4B, Table S2A).

## Microtubule destabilizing drugs disrupt the localization of PfEMP-1, KAHRP and PfTPx$_{Gl}$, leaving ACP and PfUROD localization unaffected

Based on the potential involvement of vesicles in these trafficking of PfEMP-1, KAHRP and PfTPx$_{Gl}$ but not of ACP and PfUROD, microtubules were studied; these polymers play an important role in the directional trafficking of cargo via vesicles and are also vital for the positioning of organelles such as the ER and the Golgi. Microtubule destabilizing drugs collapse the ER and the Golgi, redistributing them throughout the cell (*Cole & Lippincott-Schwartz, 1995*). A few of these drugs like vinblsatine and nocodazole were tested previously in *Plasmodium* and shown to destabilize the microtubules (*Chakrabarti et al., 2013*). Any protein dependent on vesicular transport would be expected to be dependent on microtubule integrity.

To study the role of microtubules in targeting of proteins that use different pathways, D10-ACP$_{leader}$-GFP cells were treated with the IC$_{50}$ concentration (*Chakrabarti et al., 2013*) of the microtubule destabilizing drug vinblastine. This concentration did not alter parasite morphology (Fig. S1B). To confirm that microtubule organization is indeed disrupted, immunofluorescence was performed with antibodies against tubulin. In control cells, intact microtubules forming hemispindles and sub-pellicular structures were observed (Fig. 2A). Upon treatment with vinblastine, the cells showed diffused staining throughout the cytosol (Fig. 2B). The disruption was reversed after the drug was washed out from the medium (Fig. 3). Similar results were observed for nocodazole (Fig. S5).

In control cells, PfEMP1 and KAHRP were efficiently exported out of the parasite and showed staining as seen in other reports (*Knuepfer et al., 2005*; *Wickham et al., 2001*). Upon treatment of cells with vinblastine, PfEMP1 and KAHRP showed an accumulation of both proteins in the parasite and decreased protein in the erythrocyte (Fig. 2, Fig. S2, Table S2C).

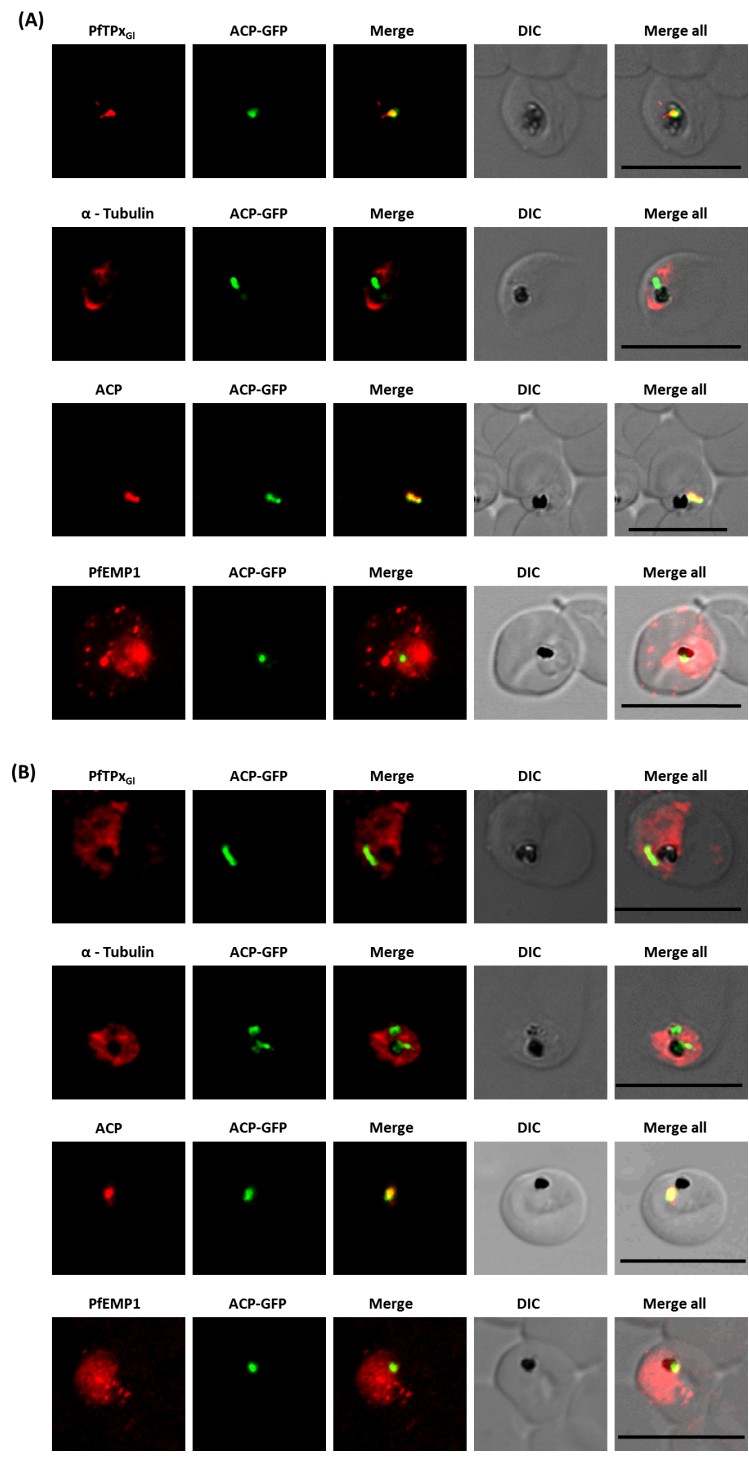

**Figure 2** Immunofluorescence images show PfTPx$_{Gl}$, microtubules, ACP-GFP, PfACP and PfEMP1 trafficking in vinblastine-treated D10-ACP$_{leader}$-GFP parasites. (A) PfTPx$_{Gl}$, microtubules, ACP-GFP, PfACP and PfEMP1 localization in solvent control parasites, (B) PfTPx$_{Gl}$, microtubules, ACP-GFP, PfACP and PfEMP1 localization in vinblastine treated parasites. (continued on next page...)

**Figure 2 (…continued)**
In these experiments, targeting to the apicoplast was inhibited in 94% of the parasites with vinblastine treatment (33 parasites counted) while an arrest of PfEMP1 was observed in 97% of the parasites (134 parasites counted). Scale Bar: 10 µm.

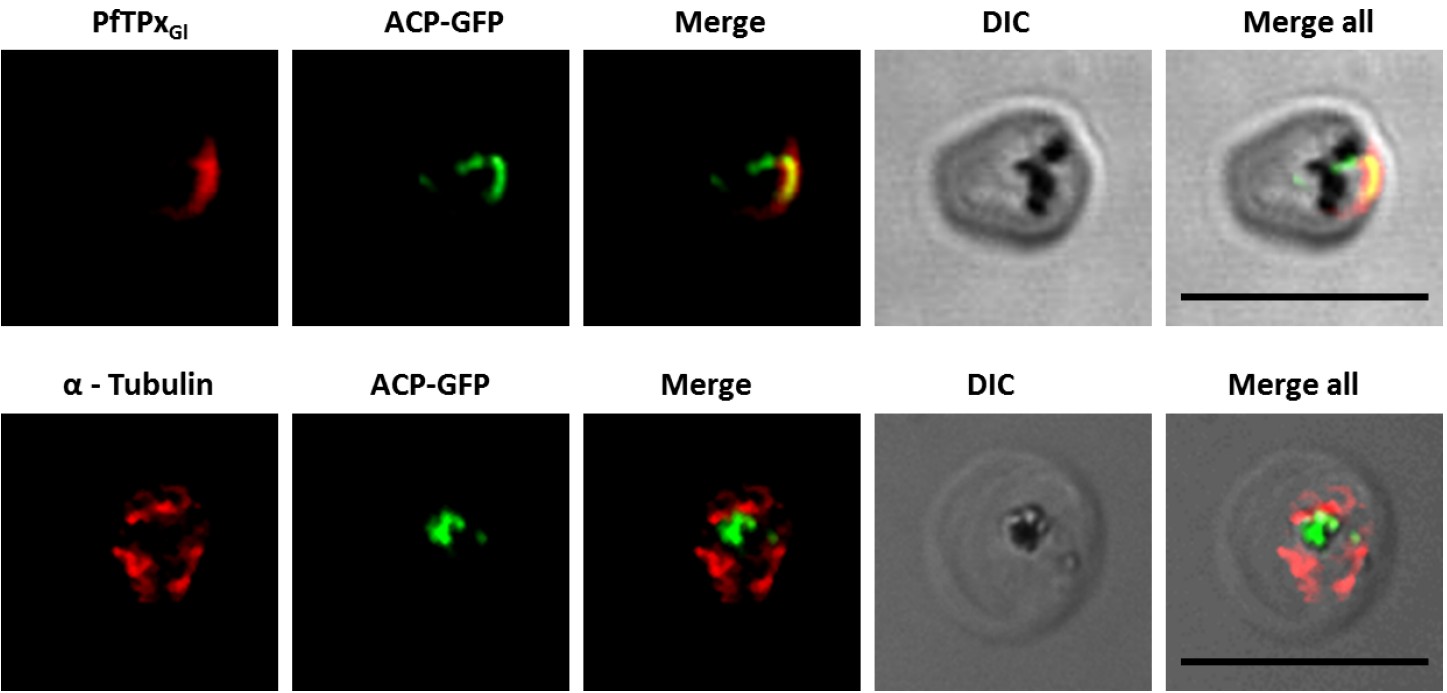

**Figure 3** **Immunofluorescence images showing PfTPx$_{Gl}$ and microtubules in D10-ACPleader-GFP parasites with drug washed out.** Reversion of PfTPx$_{Gl}$ localization to the organelles and intact microtubular structures observed in parasites in drug washed out medium after vinblastine treatment. In these experiments, localization of PfTPx$_{Gl}$ was reverted to the apicoplast in 47% parasites, while remaining 53% parasites showed mitochondrial localization (23 parasites counted). Scale Bar: 10 µm.

When PfTPx$_{Gl}$ localization was checked, the control parasites showed a staining pattern of PfTPx$_{Gl}$ (Fig. 2A) similar to untreated parasites where apicoplast localization was observed anywhere from 45% to 55% (Table S2A). When the parasites were treated with vinblastine, punctate staining of PfTPx$_{Gl}$ was visible throughout the parasite (Fig. 2B) where apicoplast localization was disrupted in more than 94% parasites. When the drug was washed out from the medium, PfTPx$_{Gl}$ was found to be co-localized with ACP-GFP within 4 h, indicating its presence in the apicoplast (Fig. 3) in 47% parasites (Table S2A).

Interestingly, in vinblastine-treated cells, ACP-GFP showed appropriate trafficking to the apicoplast (Fig. 2B) detected by a single spot. Detection of the endogenous ACP with antibodies showed the same results (Figs. 2A and 2B). Thus, the targeting of ACP is insensitive to the vinblastine that disrupts PfTPx$_{Gl}$ trafficking.

Additionally, parasites were treated with another microtubule destabilizing drug, nocodazole (at the IC$_{50}$ concentration of 17 µM). The results of ACP-GFP and PfTPx$_{Gl}$ localization after treatment with this drug closely resembled those of vinblastine treatment (Fig. S5), confirming that microtubules are involved in the trafficking of PfTPx$_{Gl}$ but not ACP-GFP to the apicoplast.

Similar to ACP-GFP, PfUROD trafficking to the apicoplast was not inhibited by vinblastine or nocodazole in 3D7 cells. However, in the same experiment, PfTPx$_{Gl}$ targeting was disrupted in the drug-treated parasites. PfTPx$_{Gl}$ targeting to the organelles was restored when the drugs were washed out from the medium, as observed by the co-localization of PfTPx$_{Gl}$ with PfUROD (Figs. S4C–S4F, Table S2A).

## Microtubule destabilizing drugs and AlF$_4^-$ do not disrupt endoplasmic reticulum morphology but have severe effects on Golgi morphology

The role of different secretory components (G proteins, small GTPases, cytoskeletal elements, and phosphatases) in maintaining the spatial distribution and structure of organelles has been shown in other eukaryotes with AlF$_4^-$ and microtubule inhibitors (*Back et al., 2004*; *Cole & Lippincott-Schwartz, 1995*). These inhibitors result in ER and Golgi dispersal, finally leading to compromised secretory traffic. However, their effects on *P. falciparum* ER and Golgi morphology are yet not known. To understand this, and assess whether the effects are consistent with the observed disruption of PfTPx$_{Gl}$ localization to the apicoplast, we treated parasites with the same concentrations of AlF$_4^-$ andvinblastine as described above, to study their effects on the ER and the Golgi. These organelles were visualized using antibodies against the ER-resident Binding immunoglobulin Protein (PfBiP) and Golgi reassembly and stacking protein (PfGRASP) respectively.

Unlike mammalian cells, no disintegration of the ER was observed in parasites subjected to vinblastine and AlF$_4^-$. Both control parasites and drug-treated parasites showed perinuclear ER morphology consistent with the normal development of the ER (Fig. 4; control parasites A, C and treated parasites B, D, Fig. S6). As expected, in experiments where parasites were treated with AlF$_4^-$ or vinblastine, PfTPx$_{Gl}$ trafficking was severely disrupted and in some cells showed partial overlap with the ER marker PfBiP suggesting an arrest in the ER. PfTPx$_{Gl}$ targeting reverted to normal when the drug was washed out from the medium (Fig. 4E).

In our analysis, staining of the parasites with anti-GRASP antibodies showed a pattern for Golgi morphology that looked like a single spot surrounded by diffuse staining for some parasites (Fig. 5) and for other parasites, a single spot with no diffuse staining (Fig. S7). This diffuse staining was different from previously published reports where a single spot was seen for the parasite Golgi (*Struck et al., 2005*; *Struck et al., 2008*). Thus, for Golgi staining the discrete spots are consistent with published data; however, a heterogenous phenotype was observed with respect to the additional diffuse staining.

In contrast to ER staining, the same treatments resulted in dispersed Golgi staining, indicating a collapse of the Golgi morphology in more than 95% parasites (Figs. 5B and 5D, Table S2B). In control parasites, the Golgi appeared to be a distinct structure as shown in Figs. 5A and 5C. In drug-treated parasites, disrupted PfTPx$_{Gl}$ showed partial co-localization with the disintegrated Golgi structures suggesting its arrest in the Golgi due to the drug treatments. Importantly, in drug was washed parasites Golgi morphology was reverted to normal in 90% of parasites observed (Fig. 5E).

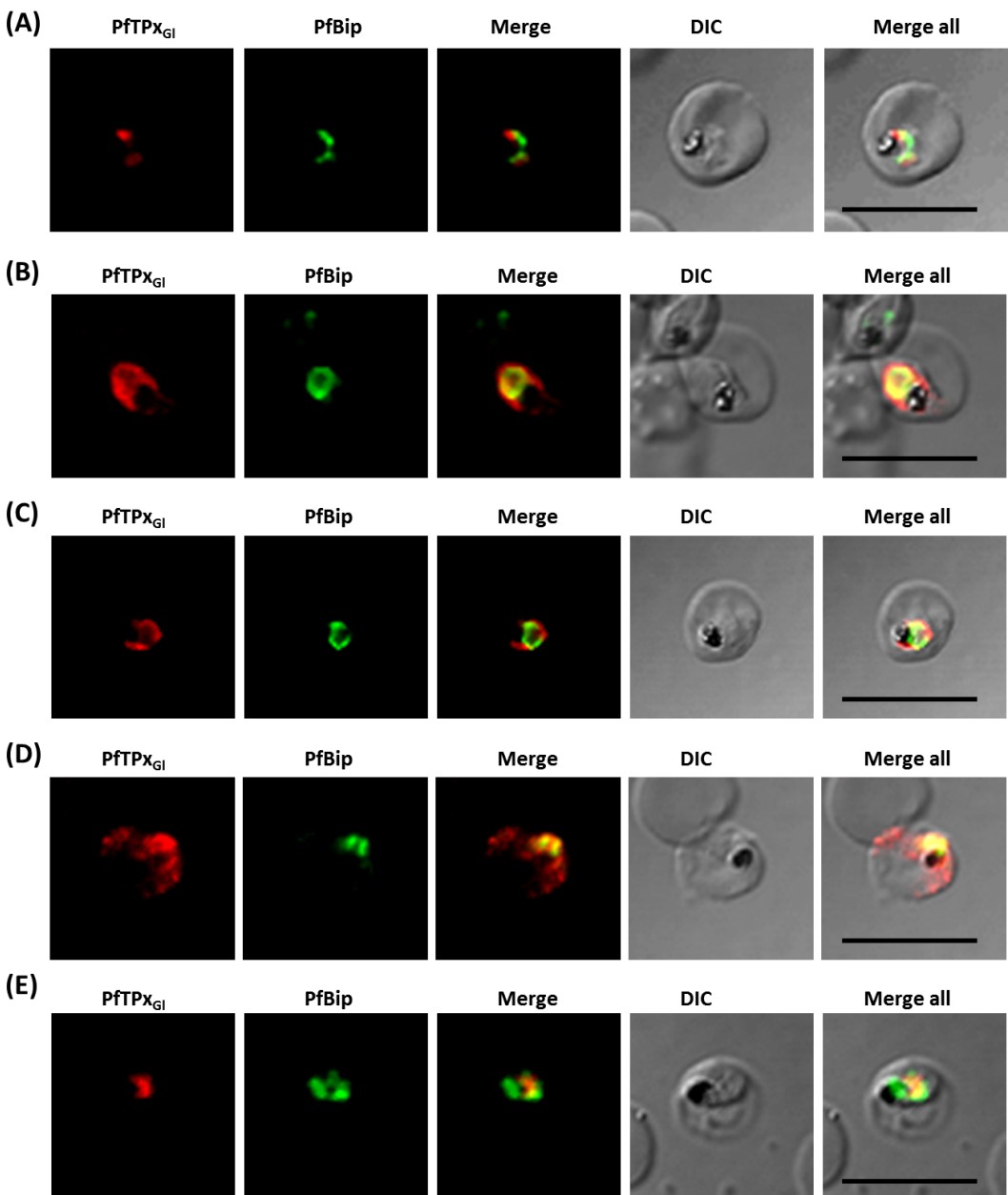

**Figure 4** **Immunofluorescence images showing the endoplasmic reticulum (ER) morphology in AlF₄⁻ and vinblastine treated parasites.** (A) PfBiP localization in control parasites for AlF₄⁻ treatment, (B) ER morphology in AlF₄⁻ - treated parasites (14 parasites were counted, none showed dispersal of ER structure), (C) PfBiP localization in solvent (PBS) control parasites for vinblastine treatment, (D) ER morphology in vinblastine-treated parasites (26 parasites were counted, none showed dispersal), (E) ER morphology in parasites reverted after vinblastine treatment (27 parasites counted). Scale Bar: 10 μm.

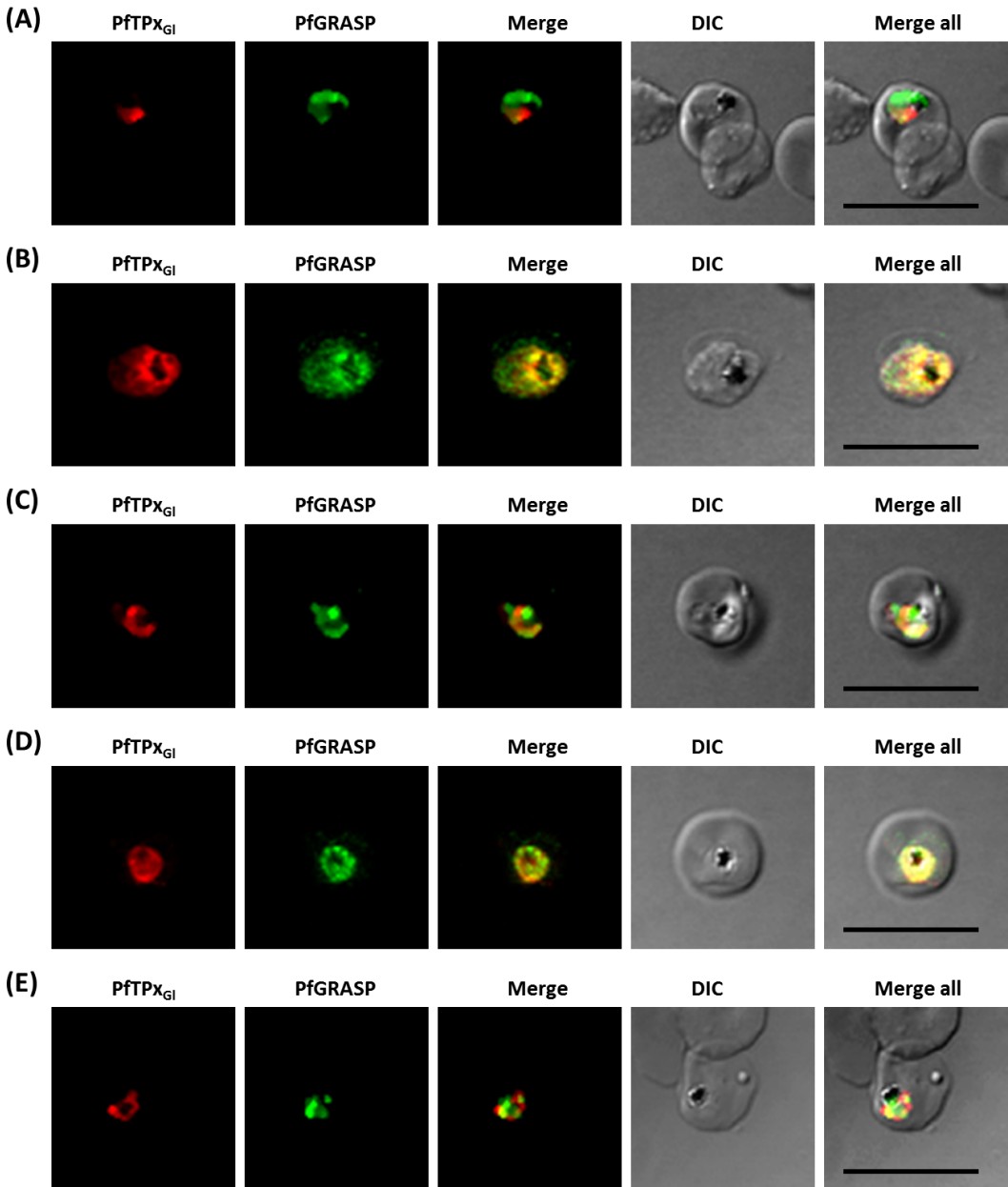

**Figure 5** **Immunofluorescence images showing the Golgi morphology in AlF$_4-$ and vinblastine treated parasites.** (A) PfGRASP localization in control parasites for AlF$_4-$ treatment, (B) Golgi morphology in AlF$_4-$- treated parasites (17 parasites counted, Golgi structure was dispersed in 95% parasites), (C) Pf-GRASP localization in solvent (PBS) control parasites for vinblastine treatment, (D) Golgi morphology in vinblastine-treated parasites (18 parasites counted, Golgi structure was dispersed in 95% parasites), (E) Golgi morphology in parasites reverted after vinblastine treatment (11 parasites counted, Intact Golgi structure was observed in 90% parasites). Scale Bar: 10 μm.

## PfTPx$_{Gl}$ is targeted to the outermost membrane of the apicoplast

Proteins located on the membranes of the apicoplast do not have conventional transit peptides. Transport of these membrane-bound apicoplast proteins from the ER to the apicoplast has been proposed to occur via vesicular trafficking (*Karnataki et al., 2007*; *Lim, Kalanon & McFadden, 2009*; *Mullin et al., 2006*). A signal anchor is thought to retain the protein in the ER membrane, following which it is targeted to the apicoplast outer membrane by vesicles (*Lim et al., 2016*; *Mullin et al., 2006*). Consistent with these observations, analysis by PlasmoAP (*Foth et al., 2003*) predicts that PfTPx$_{Gl}$ does not possess a canonical transit peptide. Interestingly, our experiments confirm that PfTPx$_{Gl}$ in parasites has a molecular weight suggestive of a lack of transit peptide cleavage (*Chaudhari, Narayan & Patankar, 2012*). These data indicate that PfTPx$_{Gl}$ might reside on the outer membrane of the apicoplast.

The association of PfTPx$_{Gl}$ with organellar membranes was investigated by differential solubilization of membranes. After hypotonic lysis, most of the apicoplast luminal protein ACP-GFP was extracted into the soluble fraction (Fig. 6A). The small fraction that was retained in the pellet was possibly due to incomplete lysis of the organelles. Unlike ACP-GFP, PfTPx$_{Gl}$ was found only in the pellet fraction indicating its association with the membrane (Fig. 6A).

To test whether PfTPx$_{Gl}$ is indeed localized on the outer apicoplast membrane, the organellar fraction isolated from D10-ACP$_{leader}$-GFP parasites was divided into two. One fraction was not subjected to fixation and permeabilization while the second fraction was fixed and permeabilized followed by immunofluorescence. ACP-GFP was used as a luminal control for permeabilization since this protein should be detected by antibodies only in permeabilized organelles. In contrast, antibodies would recognize PfTPx$_{Gl}$ in non-permeabilized organelles only if it was situated on the outermost membrane.

As expected, anti-GFP antibodies stained the apicoplast lumen (red signal) only in the permeabilized organelles, but not in the intact ones (Fig. 6C). This immunofluorescence signal colocalized with the intrinsic GFP fluorescence (green signal) of ACP-GFP. In both fractions, anti-PfTPx$_{Gl}$ antibodies clearly showed staining surrounding the fluorescent signals from luminal ACP-GFP (Fig. 6C). The halo around ACP-GFP, which was not observed for whole cells at lower magnifications, suggested that PfTPx$_{Gl}$ resides on the outermost membrane of the apicoplast. This observation is consistent with similar experiments conducted for other membrane-bound apicoplast proteins (*Kalanon, Tonkin & McFadden, 2009*; *Mullin et al., 2006*).

Additionally, the organellar fraction isolated from AlF$_{4^-}$ treated parasites was probed with antibodies against PfTPx$_{Gl}$. While the size and morphology of free intact apicoplasts (based on ACP-GFP staining) remained unaltered, no peri-organellar PfTPx$_{Gl}$ signal was observed in the immunofluorescence analysis of intact as well as permeabilized organelles, indicating the involvement of vesicles in the membrane targeting of this protein (Fig. 6D).

As PfTPx$_{Gl}$ is known to be dually localized to both the apicoplast and the mitochondrion, it was important to confirm that the PfTPxGl staining surrounding the luminal GFP signal was not mitochondrially localized protein. This was particularly important as it is known that the apicoplast and mitochondrion are closely associated in *P. falciparum*. Staining the

 

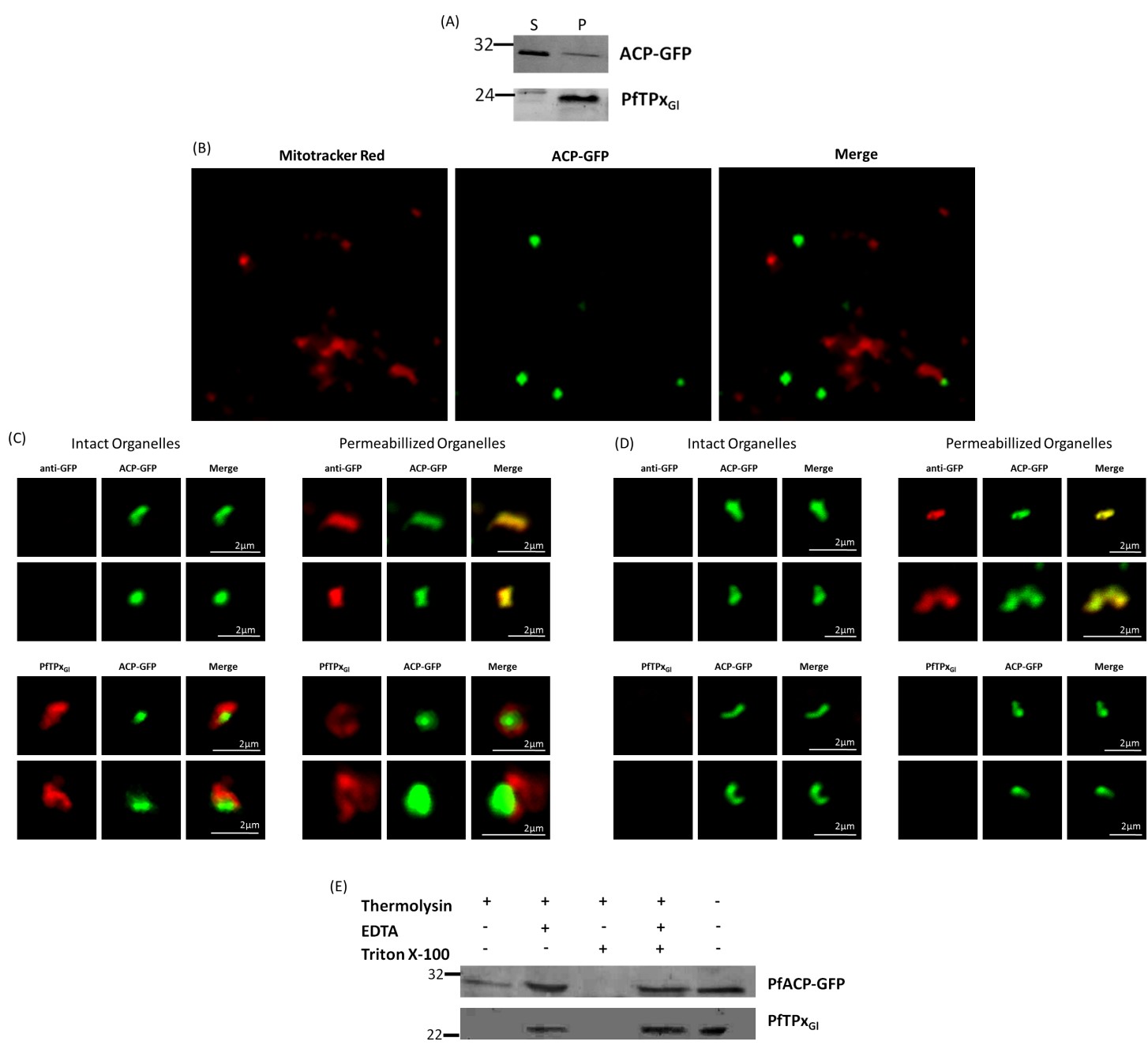

**Figure 6** **PfTPx_{Gl} localization to the outermost membrane of the apicoplast in D10-ACPleader-GFP parasites.** (A) Western blots showing association of PfTPx_{Gl} with the organellar membranes (S-Supernatant, P-Pellet) after hypotonic lysis, (B) Staining of the isolated organellar fraction with MitoTracker Red, (C) Localization of PfTPx_{Gl} to the membranes of intact/permeabilized apicoplasts from control parasites, (D) Absence of PfTPx_{Gl} in the membranes of intact/permeabilized apicoplasts from AlF_{4−} treated parasites. Scale Bars as indicated in the figures. (E) Thermolysin treatment of isolated organelles demonstrates outermost membrane localization of PfTPx_{Gl}.

organellar fraction with the red-fluorescent mitochondrial dye, MitoTracker showed that in our organellar preparations, the MitoTracker signal was clearly distinct from the ACP-GFP signal and was not seen to form a halo around the apicoplast in any microscopic field (Fig. 6B). This suggests that the PfTPx$_{Gl}$ staining surrounding the luminal ACP-GFP signal is indeed coming from the membranes of the apicoplast and not from the mitochondrion.

To support these findings, we treated the isolated organelles with thermolysin, a protease that acts outside of intact membrane compartments. Thermolysin completely digested PfTPx$_{Gl}$ showing that this protein is present on the outermost membrane. However, in these intact organelles, we found ACP-GFP largely undigested although a slightly lower amount of this protein was observed compared to controls with no thermolysin. This might be due the organellar integrity beingsomewhat compromised during handling. However, as the same experiment showed intact ACP-GFP but complete degradation of PfTPx$_{Gl}$ we infer that PfTPx$_{Gl}$ is localized on the outermost membrane.

As expected, PfTPx$_{Gl}$ was protected when the thermolysin was inhibited by the addition of EDTA. When the organelles were permeabilized by Triton X-100, both PfTPx$_{Gl}$ and ACP-GFP were digested (Fig. 6E). In conclusion, both immunofluorescence assays and thermolysin treatments strongly suggest that PfTPx$_{Gl}$ islocated on the outermost membrane of the apicoplast.

## DISCUSSION

### PfTPx$_{Gl}$ an apicoplast membrane protein, is carried by vesicles

In this report, PfTPx$_{Gl}$ is shown to be membrane-bound and appears to be on the outermost apicoplast membrane. Consistent with this data, TMPred and RHYTHM algorithms predict a trans-membrane domain in the N-terminus of this protein. Further, an analysis of the first 60 amino acids of the N-terminal leader sequence using ProtParam tool and ExPASy revealed an enrichment of hydrophobic residues (40%). Published data supports the membrane localization of the protein, as peptides corresponding to PfTPx$_{Gl}$ were found in mass spectrometric analysis of the parasite membrane proteome(*Lasonder et al., 2002*; *Yam et al., 2013*).

The membrane localization of the protein is consistent with its trafficking pathway. Transit sequences of apicoplast targeted proteins are highly enriched in positively charged residues that are recognized by organellar translocons (*Foth et al., 2003*; *Tonkin, Roos & McFadden, 2006a*). However, the N-terminus of PfTPx$_{Gl}$ has an overall negative charge indicating that it would not be recognized by translocons. We have shown in this report, that PfTPx$_{Gl}$ is an apicoplast membrane protein whose localization is disrupted by multiple inhibitors of vesicular trafficking, suggesting that this protein is trafficked to the apicoplast membrane through vesicles. The identity of these vesicles remains an area of future research.

That PfTPx$_{Gl}$ employs a vesicular pathway for apicoplast localization and also transits through the Golgi (*Chaudhari, Narayan & Patankar, 2012*) begs the question of how this protein avoids the bulk flow of protein trafficking from the Golgi through the secretory route, as seen for PfEMP-1 and KAHRP (Fig. 7). The trans-Golgi network contains an elaborate protein sorting machinery to deliver proteins to their correct destinations (*Guo,*

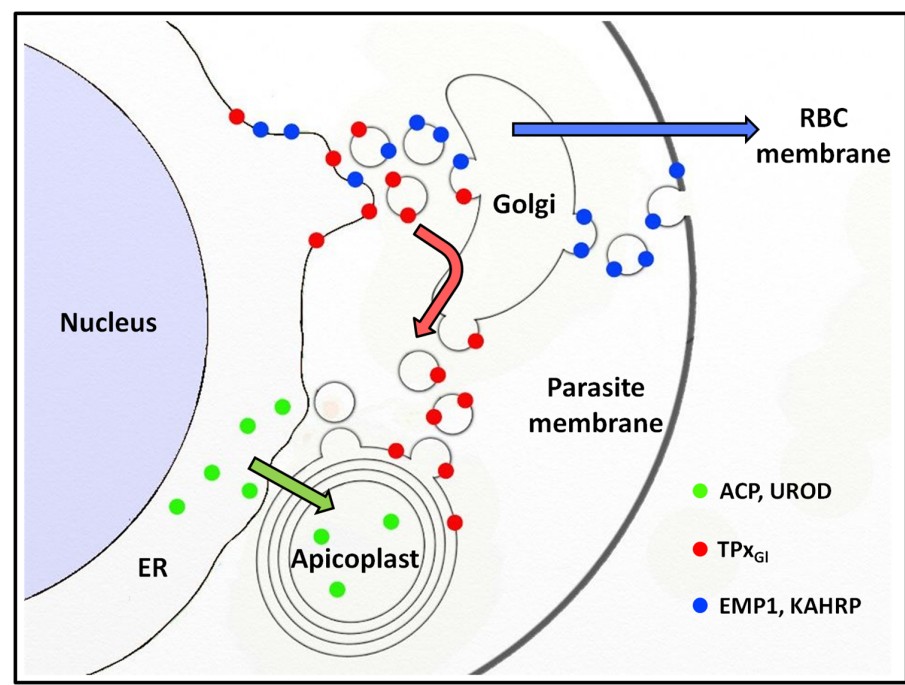

**Figure 7  Schematic representation of secretory protein targeting pathways in *Plasmodium falciparum*.** Arrows indicate the direction of the secretory protein traffic.

*Sirkis & Schekman, 2014*) and our data suggest that in *P. falciparum*, sorting to the apicoplast will be a key part of this machinery.

## Apicoplast luminal protein trafficking is independent of vesicles

Trafficking of ACP-GFP in *P. falciparum* has been shown to proceed through the ER to the apicoplast (*Tonkin et al., 2006b*) and this model has led to speculations about vesicles or direct contacts between organelles (*Kalanon & McFadden, 2010*; *Lim, Kalanon & McFadden, 2009*; *Parsons, Karnataki & Derocher, 2009*; *Tonkin, Kalanon & McFadden, 2008*). Data presented here indicates that trafficking of three luminal apicoplast proteins (ACP-GFP, endogenous ACP and PfUROD) is not inhibited by blocking the fusion of G protein-dependent vesicles. Additionally, trafficking of these proteins is insensitive to microtubule destabilizing drugs such as vinblastine and nocodazole, indicating that the microtubule tracks used for classical vesicular trafficking are not essential for their transport.

A trivial explanation for these results could be that ACP-GFP, ACP and UROD have been trafficked to the apicoplast early during the asexual cycle of the parasite (during ring stages) and are highly stable proteins with limited turnover. This would result in protein localization that is insensitive to disruption with small molecules as there is no trafficking during the time of treatment. Evidence against this possibility exists in the literature. Pulse chase experiments for ACP-GFP carried out in the late ring/early trophozoite stages show that newly synthesized, unprocessed protein can be seen at these stages (*Heiny et al., 2014*; *Tonkin et al., 2006b*; *Van Dooren et al., 2002*; *Waller et al., 2000*). Therefore, our treatments of 18 h, encompassing the rings and early trophozoites, overlap with the synthesis and trafficking

window for ACP-GFP. Similarly, Western blots of ACP-GFP after 18 h of $AlF_{4^-}$ treatment also show a fraction of unprocessed protein, similar to untreated parasites (Fig. 1C).

Based on our data, we speculate that luminal apicoplast proteins are directly trafficked from the ER to the apicoplast without G protein-coupled vesicles. That ACP-GFP may be trafficked via vesicles that are insensitive to $AlF_{4^-}$ cannot be excluded; although highly unusual, such vesicles have been reported for the endocytosis of CD94/NKG2A in natural killer cells (*Masilamani et al., 2008*). Interestingly, a recent model shows an ER-Golgi route for ACP-GFP (*Heiny et al., 2014*) which implicates vesicles in trafficking of apicoplast proteins. Data from this report suggest that, for this model too, G protein-dependent vesicles may not play a major role.

The use of microtubule destabilizing drugs has also added more insights into the unusual structure of the parasite ER and Golgi. Apart from acting as the tracks for secretory traffic, microtubules are also involved in the positioning and structural integrity of the ER and the Golgi in mammalian cells (*Cole & Lippincott-Schwartz, 1995*). However, the ER and the Golgi in *P. falciparum* appear remarkably different from those observed in mammalian cells (*Struck et al., 2005*; *Van Dooren et al., 2005*). Here, we show that disruption of microtubules in *P. falciparum* destabilizes the Golgi, while, unlike in mammalian cells, the ER does not show gross morphological changes. However, the arrest of $PfTPx_{Gl}$ in the ER and Golgi in treated cells shows that the ER function of protein trafficking through vesicles might be compromised. Interestingly, ACP-GFP localization to the apicoplast is unaffected in these experiments.

Proteins with the same destination (apicoplast) are trafficked by different routes (Fig. 7). While $PfTPx_{Gl}$ appears to be carried to the Golgi by the bulk flow of secretory traffic, ACP and UROD are diverted from this pathway, possibly by ER receptors that recognize the transit peptide (*Tonkin et al., 2006b*) which is lacking in $PfTPx_{Gl}$. Therefore, the sorting station for $PfTPx_{Gl}$ appears to be the Golgi; for ACP and UROD it appears to be the ER. It is noteworthy that differential targeting pathways for the luminal and membrane proteins of the apicoplast have been shown in related *Apicomplexan* parasite *Toxoplasma gondii* as well (*Bouchut et al., 2014*).

## CONCLUSIONS

In this study, we show that in *P. falciparum*, two different pathways exist for the localization of proteins to the apicoplast. These findings raise interesting questions regarding the molecular nature of the choices made by the parasite to direct proteins via one pathway or another (Fig. 7). Our data suggests that one of the signals may include the absence or presence of membrane anchors. A detailed understanding of these signals on proteins as well as receptors in the ER, Golgi and vesicles remain areas for future studies.

## ACKNOWLEDGEMENTS

We thank Samir Jadhav, SachinTawade, Sudesh Kumar Roy at IIT Bombay, Krishanu Ray at TIFR Mumbai for the help with confocal microscopy, Angus Bell at Trinity College Dublin for providing the marker antibodies against PfTubulin, Neel Sarovar Bhavesh at ICGEB

Delhi for providing the antibodies against PfEMP1 and PfKAHRP and Chetan Chitnis at and Pawan Malhotra at ICGEB for providing antibodies against PfBiP and PfGRASP. The *P. falciparum* D10-ACP$_{leader}$-GFP strain was obtained through MR4 (MRA-568), deposited by GI McFadden.

### Funding

This work was supported by grants from the Science and Engineering Research Board (SERB, Project File no. SB/YS/LS-354/2013) and the Board of Research in Nuclear Sciences (BRNS, Project File no. 2013/37B/18/BRNS). The funders had no role in study design, data collection and analysis, decision to publish, or preparation of the manuscript.

### Grant Disclosures

The following grant information was disclosed by the authors:
Science and Engineering Research Board: SB/YS/LS-354/2013.
Board of Research in Nuclear Sciences: 2013/37B/18/BRNS.

### Competing Interests

The authors declare there are no competing interests.

### Author Contributions

- Rahul Chaudhari conceived and designed the experiments, performed the experiments, analyzed the data, contributed reagents/materials/analysis tools, wrote the paper, prepared figures and/or tables, reviewed drafts of the paper.
- Vishakha Dey conceived and designed the experiments, performed the experiments, analyzed the data, reviewed drafts of the paper.
- Aishwarya Narayan analyzed the data, wrote the paper, prepared figures and/or tables, reviewed drafts of the paper.
- Shobhona Sharma and Swati Patankar conceived and designed the experiments, analyzed the data, contributed reagents/materials/analysis tools, wrote the paper, prepared figures and/or tables, reviewed drafts of the paper.

### Human Ethics

The following information was supplied relating to ethical approvals (i.e., approving body and any reference numbers):

The work was approved by the Institute Ethics Committee and Institute Biosafety Committee at Indian Institute of Technology Bombay. Written informed consent was provided by all the blood donors.

### Ethics

The following information was supplied relating to ethical approvals (i.e., approving body and any reference numbers):

The work was approved by the Institute Ethics Committee and Institute Biosafety Committee at Indian Institute of Technology Bombay. Written informed consent was provided by all the blood donors.

## Data Availability

The raw data has been supplied in the figures and Supplemental Files.

## Supplemental Information

Supplemental information for this article can be found online at http://dx.doi.org/10.7717/peerj.3128#supplemental-information.

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
