# Peer review of "Membrane and luminal proteins reach the apicoplast by different trafficking pathways in the malaria parasite Plasmodium falciparum"

_PeerJ, doi:10.7717/peerj.3128_

## Round 0.1 · original submission · Major Revisions

Both reviewers have raised a number of concerns that you may be able to address in a comprehensive, revised manuscript. In particular, please be sure to address the several methodological concerns raised with respect to the fluorescence images provided (especially see comments by reviewer one), including quantitation of all results where feasible and demonstration of appropriate controls. In addition, a revised manuscript should address the need for improved clarity of the overall hypothesis addressed, as well as expanded, comprehensive methods. Finally, both reviewers were concerned that the authors have overinterpreted results. A revised manuscript should demonstrate more caution and soften conclusions, especially given the lack of known specificity of the chemical inhibitors used in this study.

Reviewer 1 ·

Basic reporting

-None of the supplementary figures/tables had legends.

-Supplementary Images 6 and 7 providing the raw images related to Fig. 6 show only cut gels without clearly visible markers.

-- The title, abstract, introduction and conclusions present a somewhat discordant view of the main focus of the MS; consistency and clarity of message would help here.

Experimental design

Materials and methods are often very vague, making it difficult to assess validity of data

Image acquisition and image processing:
The authors base the majority of their arguments in this MS on immunofluorescence labeling of parasite proteins.  Unfortunately, many images appear to be full of saturated pixels. Such images can provide very misleading representations of fluorophore localization.  The authors do not describe their image acquisition procedure at all, including most crucially whether all parameters were held constant for a given label.  For instance, in Figure 3, the PfEMP1 image appears heavily saturated in the control (3A), while it does not appear saturated at all in 3B. Were these images acquired with identical settings? Were they processed identically?  

Parasite synchronisation:
The methods state “parasites were synchronized with 5% sorbitol whenever necessary” lines 109-110.  How was “necessary” determined? If the parasites were not tightly synchronized early rings at the beginning of each experiment, protein synthesized and trafficked prior to drug addition could confound analysis.  The authors need to report precisely how synchronization was done.

Determination of IC50 — highly confusing:
Figure S1A: this is referred to as a “survival assay”, with the Y axis labelled “percent death”. The methods again begin (line 127) by calling this a “survival assay”, but line 134 states “reduction in parasitemia was assessed…”, indicating that this is a straightforward growth assay, where parasitemia increases with each round of merozoite release/invasion.  Failure to grow/reinvade is not necessarily the same thing as death (that would require studies of growth/reinvasion after drug washout or a PRR type assay (i.e. PLoS One 2012, 7:e30949).  The authors should be clear about what is being measured.

IC50s are typically generated via non-linear curve fitting, whereas the data here seem to be points connected in a dotplot. More detail on IC50 calculation and providing the 95% CI would help.

Further, were these experiments done with synchronized cultures or not? Methods state (line 128) “…set-up included 2 mL of…cultures grown as described previously”, but no citation is given. If these are synchronized cultures (my assumption), why is parasitemia already increasing at 24 hours, and what does e.g. a 90% reduction in parasitemia mean at this timepoint in terms of raw parasitaemias in treated vs. controls? Some points in S1A have error bars, others appear to have none, and information on the number of biological and technical replicates was not clearly provided.  

Antibodies/Antisera:
The authors provide a Supplementary Table listing antibodies used, but it is missing crucial information. There should be an original reference (or description, if this is the initial publication) for each primary antibody/antisera used.  This is crucial for interpreting the data, as there seem to be differences in localizations from what has been previously published, e.g. for GRASP; localization reported (Struck et al 2005. doi:10.1242/jcs.02673 )was one (<16h), then later two puncta colocalized with the cis-Golgi marker ERD2 (MRA-72 from MR4) in early trophozoites. These puncta localize near protrusions of the ER as marked by BiP (MR-19 from MR4).  The GRASP localization shown in control parasites in Fig. 5 appears rather different from this.  The authors cite Siddiqui et al 2013 (lines 36-37), when describing both GRASP and BiP localizations, but this isn’t the initial description of the localization of either protein, or seemingly, of the antibodies/sera.

Validity of the findings

Major Concerns:

PfTPxGl localization needs to be rigorously quantified across parasite populations in multiple independent experiments:

According to Chaudhari et al 2012 (doi:10.1111/j.1742-4658.2012.08746.x), early trophozoites have anywhere from ~10% to ~55% of the PfTPxGl localized to the apicoplast, while late trophozoites have ~10% to ~25% of the PfTPxGl localized to the apicoplast. The same paper says that PfTPxGl can localize to the apicoplast, mitochondrion, and the parasite cytsosol, additionally stating (p.3877) that “the distribution of PfTPxGl in different asexual stages appears completely random, with no clear patterns emerging.” In this MS, the authors state in lines 85-86 that they will only focus on the apicoplast localization of PfTPxGl.  This is simply not possible given the methodology employed.

Given that Chaudhari et al. are working with immunofluorescence of fixed parasites, it cannot be known a priori that any given parasite with “disrupted” localization of PfTPxGl after treatment with AlF4 or vinblastine ever had the protein localized to the apicoplast in the first place. The example shown in e.g. Fig. 1B could simply be a parasite that had cytosolic PfTPxGl before AlF4 treatment as well.  In order to make a meaningful argument about disrupted localization of a protein that localizes randomly to the parasite cytosol, mitochondrion, and apicoplast, the authors would need to present a rigorous quantification of the various PfTPxGl localizations seen in parallel treated and control populations, as well as in pre-treatment controls.  The exact same argument holds for drug washout experiments.  Quantification of the range of protein localizations observed and the prevalence of each is necessary.

The figure legends say 87 parasites were analyzed for Fig.1 and 15 were analyzed for Fig.2 (given the random localization of PfTPxGl 87 may not be sufficient; 15 certainly is not), but for Fig.3 onwards, nothing is stated at all. The authors should always state clearly how many parasites were analyzed and the number of independent experiments.

ACP processing should be demonstrated under identical conditions to the IFAs showing ACP localization:

ACP (and ACP-GFP) undergo transit peptide cleavage during apicoplast import. The authors look at this ACP-GFP processing after AlF4 treatment in Fig 1C, but strangely, they alter their standard treatment window here. For the IFAs, rings are treated for 18 hours to draw conclusions; but for ACP-GFP processing the authors treat early trophozoites for 1, 2 or 4 hours. Allowing the parasites to grow to early trophozoites allows ample time for a pool of processed ACP-GFP to accumulate in the apicoplast, and such short AlF4 treatment windows may not alter the balance between processed/unprocessed forms. ACP (or ACP-GFP) processing needs to be shown after an identical treatment to that used for the IFAs.

Other points:

- Figure S2 shows a doubly-infected RBC for the KAHRP control, and singly infected RBCs for the two treatments. In addition to doubts on image acquisition explained above, protein localization should ideally be compared in RBCs with the same multiplicity of infection. This is absolutely critical when assessing proteins exported to the RBC; a doubly infected RBC will of course show more KAHRP labelling in the RBC than a singly infected one.

- Images in several figures were clearly acquired with different magnifications based on the size of RBCs, yet sometimes only one scale bar is shown per figure/panel (e.g. Figure 2B). Either all images need to be at the same magnification, or scale bars should be provided for each set of parasite images.

- It would be very helpful to include images of DNA using e.g. Hoechst for each set of images. Indeed, the authors refer to the “perinuclear ER morphology” (lines 339-40) of the ER marker BiP in the control parasites (Fig. 4), but there are no nuclei to be seen.  

- Fig. 2: The ACP-GFP compartment appears quite different, and not in a consistent way, in the vinblastine-treated images vs. the controls.  Why?

- Fig. 6: There appears to be substantially less ACP-GFP in the thermolysis-treated organelles in Fig. 6E than those without thermolysin. ACP-GFP levels likewise greater when organelles are treated with thermolysin + EDTA than thermolysin alone. How do the authors explain this?

Reviewer 2 ·

Basic reporting

- Abstract/ Lines 81-82: The secretory pathway in P. falciparum has more destination compartments than the 3 mentioned. This should be made clear.
-Lines 70-75 – These statements oversimplify the prior data, and more specific language should be use. Neither paper “showed” direct ER trafficking or trafficking via the Golgi, they more “suggested” Golgi-independent or -dependent trafficking based on indirect assays. Overall, the question of whether the Golgi is involved in trafficking to the apicoplast is still open and cannot be conclusively decided from either paper, and the authors should make it clear that this aspect of trafficking to the apicoplast is one that is still poorly understood.
-Lines 241-242 – Again more specific language is suggested. The Chaudhari 2012 paper showed that TPxGl trafficking to the plastid was disrupted upon Brefeldin A (BFA) treatment. While this suggests Golgi-dependent trafficking, it does not conclusively show trafficking via the Golgi since loss of localization of TPxGl could be a secondary effect of Golgi disruption. Bc the targeting of TPxGI and ACP are affected differently by brefeldin, there is already pre-exisiting data that their targeting mechanisms are different.
-Lines 76-80 – The authors should also mention that other groups have localized TPxGl to both the cytosol and the apicoplast (Kehr et al. (2010) PLOS Pathogens 6:e1001242). While it is possible that the cytosolic localization in Kehr et al. is an artifact due to ectopic expression of a GFP reporter, this reported localization is still relevant to interpreting the data presented here.
-Lines 87-89 – The drug effects described have only been characterized in other organisms and need to be clearly stated so. Some caution should be taken that the drugs may not have target the same functions in Plasmodium.
- Line 216-229: This paragraph does not clearly fit into the standard format of a results section as it contains methodological details unconnected to any specific results.
-Line 304 – The reference to Figure 3 in this line seems incorrect, should this be referring to Figure 2?
-Line 375 – Figure reference should be “6C” not just “C”
-Missng citation. The data presented here are consistent with data from Toxoplasma that also suggests that lumenal and outer membrane apicoplast proteins traffic via different mechanisms: Bouchut et al. (2014) PLOS ONE 9:e112096. The Bouchut paper should be referenced here.

Experimental design

- The “knowledge gap” being addressed by the study is not well-defined and consistent in the title, abstract, and introduction. Are the authors trying to provide evidence whether luminal apicoplast proteins are transported via the Golgi? Is the main question whether apicoplast proteins with N-terminal targeting sequences and those without targeting sequences are transported via different pathways? Both are relevant and meaningful questions, however needs to be clearly stated which is the point of the paper.
- Lines 325-349/ Figures 4+5: The specificity, mechanism-of-action, and effects on secretory pathway of the drugs used has not been clearly demonstrated in Plasmodium (as far as we are aware). Any available evidence for drug effects in Plasmodium should be referenced. This results section describing the effects of AlF4- and vinblastine on the ER and Golgi should also be moved to the beginning of the results section as the first results paragraph. A clear understanding of the drug effects in Plasmodium are necessary in order to interpret other results that use these drugs as a tool to decipher apicoplast transport pathways.
- Lines 338-343/ Figure 4: BiP localization is usually described as perinuclear, so it is difficult to interpret these images without nuclear staining. If the authors have nuclear images that were taken alongside the ones presented, it would greatly enhance the ability of the data to be interpreted. In general, any description of ER localization would benefit from having nuclear co-staining.
- In general for all images, the study would be improved if microscopy data were quantified so that heterogeneity within the population can be assessed. Particularly because the drugs are used at 50% effective doses, it is expected that the cellular changes may not be fully penetrant in the population.
-Lines 233-235: The authors should make it clear that past works that described the molecular target of AlF4- were deduced from experiments not conducted in Plasmodium parasites. While it is possible that this drug has a similar target in Plasmodium and other eukaryotic cells, the target of this drug has not been shown in parasites. This is an important caveat that should be mentioned.
- Figures 3, 4E, 5E, and some supplemental: For drug washout experiments, if only treating at EC50, how can you know that parasites analyzed after washouts are actually parasites that were inhibited and then recovered as opposed to parasites that were simply never affected by the drug in the first place? Without either treating with saturating drug or following a single cell over time, I find it difficult to interpret the washout data. Again quantification of the image data may help assess potential heterogeneity and phenotypic changes with washout.

Validity of the findings

-The authors convincingly show that treatment of parasites with AlF4- and vinblastine cause different phenotypes for proteins targeted via different mechanisms: exported proteins and outer membrane associated apicoplast proteins that are hypothesized to traffic via a membrane anchor are disrupted, whereas both endogenous and ectopic apicoplast proteins containing the traditional signal sequence-transit peptide are unaffected. This strongly suggests that outer membrane vs. lumenal apicoplast proteins traffic via different signals and routes and require different molecular machinery.
-The authors also convincingly show that TPxGl is membrane-associated and localized to the outermost membrane of the apicoplast. To my knowledge, this is only the second outer membrane-associated protein identified in Plasmodium and the fourth in Apicomplexa overall. How trafficking of outer membrane versus lumenal apicoplast proteins occurs is a very open question still, and the data presented here are a step towards understanding the differences in these two important targeting pathways.

-The authors call PfTPxGl a membrane protein in the title (and elsewhere in the manuscript) implying it is an integral membrane protein, but this has not been sufficiently demonstrated. Whether TPxGl is truly a membrane protein or is just peripherally associated with the membrane is still unclear.

Lines 419-421, 429-458: These statements in the discussion and others like it in the abstract, and introduction are beyond what can be directly concluded from the data and more in realm of speculation and should be identified as such. Treatment with these drugs may be off-target and are likely to have pleiotropic effects on the cells, making it difficult to conclude that the observed phenotypes are directly due to disruption of TPxGl-carrying vesicles. However, the data do very strongly suggest that ACP and TPxGl target through different mechanisms as they are affected differently by chemical perturbation.

---

## Round 0.2 · Major Revisions

The revised manuscript was carefully re-reviewed, and, as you will see, the reviewer continues to have substantial concerns about the image processing and the quality of the data. The reviewer has provided specific and thoughtful advice for improving the quality of image processing and data. Please note that your revised manuscript must directly address these concerns about experimental rigor, before it can be accepted for publication.

Reviewer 1 ·

Basic reporting

see 2.

Experimental design

I thank the authors for the work done to improve the manuscript. There is likely a nice story to be uncovered here, but I am not convinced by the clarifications provided and the revised manuscript that all investigation has been performed rigorously and to a high technical standard.

Most importantly, since the authors have clarified that all parameters were held constant for a given label during image acquisition, the amount of differential post-processing that was performed on individual panels that are being directly compared does not provide confidence. Figure 2 (mis-identified as Figure 3 in my initial review) which seemed to show PfEMP1 images that had been acquired and/or processed in non-equivalent ways was scrutinized in ImageJ in comparison to a published PNG file (Fig. 1 from dx.doi.org/10.1371/journal.ppat.1000328 — to which I have no affiliation whatsoever— referred to as ppathFig1) that shows similar multichannel immunofluorescence data comparing localization of Plasmodium proteins in infected RBCs. A PDF showing the original PNGs (ppathFig1 vs. Fig2 of this MS) and subsequent matched, linear adjustments in ImageJ side by side has been provided to the editor. ppathFig1 shows the regular behaviour amongst all panels that would be expected for images acquired and processed similarly, while Fig. 2 from this MS has a number of inconsistencies.

Import both files into ImageJ and consider only the fluorescence panels. Rescaling pixel intensities (Image/Adjust/Brightness - Contrast) to a min and max of 0 results in all panels of ppathFig1 turning a uniform white, while it generates a complete hodgepodge in Fig. 2 of this MS. Some fluorescent images saturate to white (2A PfEMP1 single channel and merge), while other saturate fully to colors (2A row 3). Some do not saturate fully, but show clear digital noise (e.g. ACP-GFP in 1A row 1) while others do not present any digital noise at all (e.g. ACP-GFP in 1B row 1). PfEMP1 intensities in Fig. 1B (panel4) have clearly been adjusted differently in the single channel and merged image. Further, some but not all ACP-GFP “green” single channel images, (pseudocolored obviously) show acquisition of substantial “red” pixels upon rescaling (1A row 1,2 and 1B Row 3). The single channel ACP-GFP image in 1A appears to have substatially more yellow pixels in the single channel than in the merge. Rescale the max pixel intensity to 7 (leave the min at 0), and you can see more clearly the quasi-uniform behaviour of all images in ppathFig1 at a step before all panels are white; in comparison all the same features described can already be seen in Fig. 2 from this MS. Whether this type of differential manipulation (and manipulations may well be completely linear) is done directly at the microscope or using ImageJ or the like after image acquisition does not really matter. The result is the same—images that are not directly comparable between experiment and control, or even between parasites in the same treatment group with a single label (e.g. ACP-GFP).

I’ve illustrated this in depth as I hope this is just out-of-control image processing, trying to over-accentuate features that are indeed there. I would strongly recommend returning to the original images, reconfirming that truly all imaging parameters were held constant (check the metadata!), and then performing careful and largly consistent manipulations where essential. Alternatively, images that illustrate differential protein localization after compound treatments may be best acquired in duplicate, with one set of images holding all parameters constant at a point so that no saturated pixels are present in the vehicle control, and then all other treatments are imaged with those identical settings, followed by imaging with the gain alone adjusted (all other parameters are held constant) so that each image best represents the localization of a fluorophore in that particular cell.

I have not scrutizined the others figures, but the authors should. Additionally, there are still shortcomings in this MS as identified in the original review. As there are are no word limitations being imposed here, the authors should take seriously the journal's instruction, "methods described with sufficient detail and information to replicate", and not hesitate to engage thoughtfully and critically (of their own data, as well as that of others) with relevant literature in the discussion. For example, identification of the Golgi. See Struck et al. 2005 Fig.5 for anti-grasp localization in comparison to Grasp-GFP; details of the mouse anti-grasp mouse sera are provided in Materials and Methods of that paper. The specific details of the antibodies/sera used (e.g. raised against a peptide? Which peptide? Recombinant full-length protein? Specificity determined how?) are always germane to analysis and interpretation of the data. The authors still fail to provide meaningful references or primary information about several antibodies that this MS relies on, including the anti-grasp which provides a rather different pattern of cis-Golgi staining than what has been previously reported for well-described reagents, including anti-ERD2, anti-Grasp, and Grasp-GFP (search the MR4 reference in original review for ERD2). This may again be down to imaging (or image processing), but if not, should be discussed openly, at a minimum, so that the reader can critically assess the data / conclusions of this MS WRT other works.

Validity of the findings

See 2.

Annotated reviews are not available for download in order to protect the identity of reviewers who chose to remain anonymous.

---

## Round 0.3 · accepted · Accept

Thank you for your persistence in improving your work and your careful response to the reviewers. Nicely done.

Reviewer 1 ·

Basic reporting

The authors have improved the MS, and addressed previous points of concern.

Experimental design

See 1.

Validity of the findings

See 1.